# Fabrication of Millable Polyurethane Elastomer/Eucommia Ulmoides Rubber Composites with Superior Sound Absorption Performance

**DOI:** 10.3390/ma14237487

**Published:** 2021-12-06

**Authors:** Yuhang Dong, Dexian Yin, Linhui Deng, Renwei Cao, Shikai Hu, Xiuying Zhao, Li Liu

**Affiliations:** 1Key Laboratory of Beijing City on Preparation and Processing of Novel Polymer Materials, School of Materials Science and Engineering, Beijing University of Chemical Technology, Beijing 100029, China; dyjdyh1881030@163.com (Y.D.); yindx96@163.com (D.Y.); denglh1107@163.com (L.D.); renweicao@163.com (R.C.); 2Beijing Engineering Research Center of Advanced Elastomers, School of Materials Science and Engineering, Beijing University of Chemical Technology, Beijing 100029, China

**Keywords:** eucommia ulmoides rubber, millable polyurethane elastomer, damping, sound absorption

## Abstract

Sound absorbing materials combining millable polyurethane elastomer (MPU) and eucommia ulmoides rubber (EUG) were successfully fabricated via a physical blending process of EUG and MPU. The microstructure, crystallization performances, damping, mechanical and sound absorption properties of the prepared MPU/EUG composites were investigated systematically. The microstructure surface of various MPU/EUG composites became rough and cracked by the gradual incorporation of EUG, resulting in a deteriorated compatibility between EUG and MPU. With the increase of EUG content, the storage modulus (E’) of various MPU/EUG composites increased in a temperature range of −50 °C to 40 °C and their loss factor (tanδ) decreased significantly, including a reduction of the tanδ of MPU/EUG (70/30) composites from 0.79 to 0.64. Specifically, the addition of EUG sharply improved the sound absorption performances of various MPU/EUG composites in a frequency range of 4.5 kHz–8 kHz. Compared with that of pure MPU, the sound absorption coefficient of the MPU/EUG (70/30) composite increased 52.2% at a pressure of 0.1 MPa and 16.8% at a pressure of 4 MPa, indicating its outstanding sound absorption properties.

## 1. Introduction

With the rapid development of modern industries, there are an increasing number of vibration tools and high-power mechanical devices with strong vibrations and high levels of noise pollution, both of which seriously affect not only the normal operation of electronic devices and instruments but also human health [1,2,3,4]. To solve the aforementioned problems, many damping materials have been designed and fabricated by scientists and engineers [5,6]. Among these materials, rubber damping material with unique viscoelasticity, which could reduce vibration and noise pollution via transforming sound energy or vibration energy into heat energy, has received enormous academic and industrial interest [7,8]. In recent years, increased social demands require rubber damping materials to maintain their damping capacity under pressure. However, traditional rubber damping materials such as styrene butadiene rubber (SBR) and polyurethane (PU) tend to deform under high pressure on account of their low modulus, resulting in the reduction of their vibration reduction and sound absorption performances [9]. Therefore, it is of great importance to research new types of rubber damping materials to satisfy modern social demands.

Eucommia ulmoides is a relict plant deposited by quaternary glaciers. The bark, leaves, fruits and seeds of eucommia ulmoides all contain eucommia ulmoides rubber (EUG), which is a type of bio-based polymer material [10,11]. EUG, an isomer of natural rubber (NR), crystallizes at room temperature by the ordered trans arrangement. EUG displays two forms of crystals (α and β) and owns two loss peaks, corresponding to the glass transition and crystallization melt transition respectively [12]. EUG offers a higher modulus than traditional rubber owing to its crystallization performance, which could resist deformation under high pressure [13]. Blending or modifying EUG with other synthetic rubbers or plastics could fabricate a variety of rubber products with excellent comprehensive properties via regulating their degree of crosslinking [13,14]. For instance, Zhang et al. [15] researched the sound absorption performance of EUG and its conventional damping rubber blends systematically, finding that an equivalent amount addition of EUG could effectively improve the sound absorption performance of chloroprene rubber (CR) and chlorobutyl rubber (CIIR) in the low-frequency region. Moreover, when the mechanism of sound absorption of EUG was examined, it was indicated that when the sound wave of the incident reached the sound absorbing materials, EUG crystals played a key role in sound wave conversion. Specifically, the incident compressional wave was converted into the incident shear wave at the interface between the hard phase (crystalline region) and soft phase (rubber region), causing the shear deformation of the materials and increasing the loss of sound energy. The above research shows that EUG has a lot of potential in the field of sound absorption.

Polyurethane elastomers, the products of the addition polymerization reaction of polyester (polyether), isocyanate and low-molecular-weight dihydric alcohol (as chain extender), are also commonly used for preparing sound absorbing materials [16,17]. Depending on different processing methods, polyurethane can be divided into casting polyurethane elastomer (CPU) [18], thermoplastic polyurethane elastomer (TPU) [19] and millable polyurethane elastomer (MPU) [20]. Compared with other polyurethanes, MPU displays a lower molecular weight and fewer branched chains and it presents transparent rubber at room temperature [21,22]. Furthermore, MPU can be processed via traditional rubber machines and technologies used to plastify, mix and vulcanize. More importantly, the vulcanization speed of MPU can be changed by adjusting the vulcanization temperature, and the vulcanized MPU cam exhibit good mechanical properties, excellent water resistance, wear resistance and decay resistance properties as well as favorable sound absorption performance [23,24].

In this research, various MPU/EUG composites with a good damping performance were fabricated using a physical blending method, and the relationship between the microstructure of MPU/EUG composites and their macroscopic performance (including dynamic and static mechanical properties as well as sound absorption properties) was investigated systematically at different ratios of MPU and EUG. The sound attenuation performance of MPU/EUG composites was improved thanks to the crystallization of EUG. Compared with pure MPU material, the MPU/EUG composites exhibited improved sound absorption performance under high pressure, which provides a promising strategy for the development of new types of rubber damping materials.

## 2. Experimental

### 2.1. Materials

EUG was produced by Shandong Beilong Eucommia ulmoides biomaterial Co., Ltd. (Shandong, China). MPU (SKEU851-4035) and surfactant (NH-2) were obtained from Nanjing Sikai Rubber and Plastic Co., Ltd. (Jiangsu, China). The MPU (SKEU851-4035) used in this work is a M type unsaturated polyether rubber with good wear and hydrolysis resistance. Other experimental materials were bought from common manufacturers in the rubber industry.

### 2.2. Preparation of Various MPU/EUG Composites

Various MPU/EUG composites were prepared using a three-stage method, and the experimental formula is shown in Table 1. First, MPU was plasticized at room temperature on the open mill. After wrapping the roll properly, stearic acid (SA) and activator (NH-2) were added, after which accelerants (D and DM) and sulfur were added for uniform mixing. After repeated rubber cutting and uniform mixing of MPU as well as other materials, the roller spacing was adjusted to the minimum and then a triangle bag shape was made three times. The above steps were repeated once before the composites were removed.

The hot roll mill was heated to 80 °C, and EUG was put into the mill for plasticizing into a black gel shape, before the roll was wrapped properly. Then, zinc oxide, SA, accelerant (DZ) and sulfur were mixed respectively. The roller spacing was then adjusted to the minimum, and triangle bag shape was made three times. The composites were taken out after the uniform mixing of MPU and other materials. Finally, the prepared MPU and EUG were blended to fabricate diverse MPU/EUG composites with the ratio of 100/0, 90/10, 80/20, 70/30, 60/40, 50/50 and 0/100 on the hot roll mixer (Wuxi Sanjiang Machinery Co., Ltd., Jiangsu, China) (as shown in Figure 1a).

The vulcanization curves with the burning time (T_10_) and the curing time (T_90_) of various MPU/EUG composites were obtained via a vulcanizing instrument (displayed in Figure 1b). About 40 g of diverse MPU/EUG composites which has been placed for 12 h were tested at a vulcanization temperature of 150 °C to obtain T_90_. The vulcanized MPU/EUG composites were prepared by an electric plate vulcanizing machine (Hangzhou Dongfang Machinery Co., Ltd., Zhejiang, China) at 15 MPa for T_90_+2 min.

### 2.3. Characterizations

#### 2.3.1. SEM Observation

The fracture surface morphology of diverse MPU/EUG composites was examined via SEM (S-4800, Hitachi, Japan). The width and thickness of prepared composites were 1 and 2 mm respectively. Before observations, diverse MPU/EUG composites were freeze-fractured after being immersed in liquid nitrogen to obtain a good fractured surface and were then sputter-coated with Au (shown in Figure 2a).

#### 2.3.2. Differential Scanning Calorimetry (DSC)

The thermal properties of diverse MPU/EUG composites were examined by DSC (STARe system, Mettler Toledo, Switzerland). Different composites were first quickly heated from room temperature to 100 °C, held for 5 min to remove thermal history, then cooled to −100 °C with a rate of 10 K/min and then heated back to 150 °C at the same rate. The crystallinity (*X_c_*) of diverse MPU/EUG composites was calculated via Equation (1):(1)$$X_{c}=\frac{\Delta H}{\Delta H_{0}}\times 100\%$$
in which Δ𝐻 was the melting enthalpy, Δ𝐻_0_ was the standard melting enthalpy of pure EUG crystallized 100% (186.8 J g^−1^).

#### 2.3.3. X-ray Diffraction (XRD)

XRD analysis was employed to test the crystal forms of diverse MPU/EUG composites with an X-ray diffractometer (D/max-2500VB2PC, Rigaku, Japan) at a scattering range of 5–30° and a scanning rate of 3° min^−1^. The test voltage and test current were 40 KV and 100 mA respectively.

#### 2.3.4. Dynamic Mechanical Analysis (DMA)

DMA was performed using a dynamic mechanical thermal analyzer (DMA; VA3000, Metravib, France). Diverse MPU/EUG composites were cut into rectangles (10 mm × 6 mm × 2 mm). The composites were first cooled from 25 °C to 100 °C at a rate of 10 K/min, held for 10 min and then heated to −100 °C at a rate of 3 K/min at a frequency of 10 Hz under a deformation of 0.1%.

#### 2.3.5. Sound Absorption Coefficients Performances

The sound absorption coefficients of diverse MPU/EUG composites were tested by an acoustic tube measuring system (diameter = 120 mm, height = 50 mm) at a temperature of 22 °C, water temperature of 15 °C and humidity of 52% RH under the frequency bands of 3–8 kHz and pressure bands of 0.1–4 MPa.

#### 2.3.6. Mechanical Properties

The mechanical property tests of diverse MPU/EUG composites were examined by an electronic tensile tester (in Figure 2b) (AI-7000S1, High-speed Railway Testing Instrument, Guangdong, China) according to the GB/T 528-2009 standard [25]. The dumbbell-shaped composites (3.2 mm × 2 mm) were stretched at 500 mm/min. The hardness of the prepared composites was measured by a durometer (SAC-J, Ruidayu Instrument, Beijing, China) according to the GB/T531.1-2008 standard [26].

## 3. Results

### 3.1. Scanning Electron Microscopy (SEM) Observation

SEM was utilized to explore the effect of EUG content on the microstructure of the MPU matrix. Various MPU/EUG composites with the ratio of 100/0, 90/10, 80/20, 70/30, 60/40 and 50/50 are shown in Figure 3a–f, respectively. As can be seen from Figure 3a, the fracture morphology of pure MPU was smooth and flat, and the molecular chain orientation caused by the stress during the composite preparation can also be observed in Figure 3a. With the addition of EUG content, the fracture morphology of various MPU/EUG composites became increasingly rough, and some larger particles appeared gradually, which was actually the crystalline regions of the EUG matrix [27,28]. In the matrix of various MPU/EUG composites, MPU is mainly cross-linked with EUG by the double bonds in its side chains and the number of crosslink sites was relatively low. As a result, uncross-linked EUG crystallized in the MPU/EUG matrix [10]. Meanwhile, the presence of side chains enhanced the polarity of MPU, which deteriorated the compatibility of MPU and EUG in the MPU/EUG matrix. With an increase in EUG content, therefore, cracks and the voids (caused by EUG crystallization shedding) increased gradually on the microstructure of various MPU/EUG composites.

### 3.2. Crystallization and Melting Properties

The influence of EUG content on the crystallization and melting performances of various MPU/EUG composites was investigated, and the related data are displayed in Figure 4 and Table 2. In Figure 4a, it can be observed that at about −55 °C curvilinear steps were caused by the glass transition of the composites and the curvilinear steps became less distinct with the gradual addition of EUG. This is mainly because at the glass transition temperature, a large number of EUG crystals still existed in the composites owing to the high melting point of the EUG crystalline region, which hindered the molecular chain movement of the composites [29,30]. Besides, it can be found in Table 2 and Figure 4 that with the increase of EUG content, the crystallinity of various MPU/EUG composites gradually increased and their crystallization melting peak gradual appeared and became more and more obvious at about 40 °C [28]. It is well-known that there are α and β crystal types in the EUG crystalline region, but only one crystallization melting peak can be observed from Figure 4a.

To explore whether there were two crystal types in various MPU/EUG composites, XRD tests were employed to further investigate the crystallization properties of the composites, and their XRD curves in the scanning range of 5–30° are displayed in Figure 4b. As shown in Figure 4b, the crystallization peak of EUG appeared at 2θ = 19.2° and 2θ = 23.2° when the MPU/EUG ratio was 70/30, and the crystallization peak became more and more obvious with the increase of EUG content. However, the α-type crystallization peak of EUG was mainly at 2θ = 26.6°, while no crystallization peak appeared at about 2θ = 26.6° (as shown in Figure 4b) [31,32], proving that the crystal type of EUG in various MPU/EUG composites is mainly β-type crystals. In reality, the forming of α-type crystals was not only related to the crystallization temperature but was also affected by the heating rate, so the forming conditions of α-type crystals were stricter than that of β-type crystals [33,34]. In contrast, the crystallization temperature of β-type crystals was relatively low and its forming conditions were simple [33,35]; hence, the existed crystals in the MPU/EUG composites were mainly β-type crystals.

### 3.3. Damping Properties

In the context of sound absorbing materials utilized under vibrating conditions, their dynamic mechanical properties can better reflect their performances in practical applications. Therefore, DMTA was employed to investigate the dynamic mechanical properties of various MPU/EUG composites, and the relative results are shown in Figure 5. As can be seen from the tanδ–T curves in Figure 5a, the damping loss peaks of pure MPU and pure EUG were about −30 °C and −50 °C respectively, and another loss peak which appeared at 40 °C was mainly caused by the crystallization melting process of EUG [36]. With the increase of EUG content in various MPU/EUG composites, the peak value of their tanδ decreased and the damping temperature range (tanδ > 0.3) gradually became narrow. Compared with that of pure MPU, the tanδ peak value of MPU/EUG (70/30) composite decreased from 0.79 to 0.64. This was mainly because the molecular chain structure of EUG displayed trans-chain and relative regular structures. With the increase of EUG content, the trans-chain structures increased in MPU/EUG composites and the internal friction decreased [10,37], resulting in poor damping performance. Similar results can also be found in the literature [15,38].

From the storage modulus (E’)-T curves in Figure 5b, it can be observed that the E’ of various MPU/EUG composites displayed a plateau in the temperature range of −50–40 °C. Adding 10 content of EUG into MPU did not change the E’ of MPU/EUG composite, but with the continued increase of EUG content, the E’ of various MPU/EUG composites clearly increased from 10^6^ Pa to 10^8^ Pa, indicating that the E’ of various MPU/EUG composites can be regulated by changing the proportion of EUG in the composites. The crystals in EUG offered high-modulus in the MPU/EUG matrix, which could explain the above phenomena [10,39,40,41]. Moreover, the E’ of various MPU/EUG composites significantly decreased with the increase of EUG content when the temperature was raised above 40 °C, owing to the crystals melting in various MPU/EUG composites. The applicable temperature of sound absorption materials was about 0–30 °C, so the prepared MPU/EUG composites were able to achieve impedance matching with water by adjusting the ratio of MPU/EUG (as well as E’) within this temperature range, thereby improving sound absorption performance.

### 3.4. Sound Absorption Performances

Figure 6 displays the sound absorption coefficient curves of various MPU/EUG composites under different pressures. As shown in Figure 6, the sound absorption data is quite chaotic in the frequency range of 3–4.5 kHz, which might be caused by the instability of the instrument in this frequency domain. In the frequency range of 4.5–8 kHz, the sound absorption coefficient of various MPU/EUG composites increased gradually with the addition of EUG, and the more EUG added into MPU, the more obvious the sound absorption coefficient improved. There are two reasons that account for the above phenomenon. On the one hand, the small EUG crystals in MPU matrix could diffuse the sound waves incident into various MPU/EUG composites, changing the propagation path of sound waves and dissipating more acoustic energy [15,41,42,43]. On the other hand, the addition of EUG sharply improved the modulus of various MPU/EUG composites because of the high modulus of EUG, accelerating the propagation speeds of sound waves in the composites and achieving impedance matching with water [44,45], thus improving the sound absorption performance of various MPU/EUG composites.

The average sound absorption coefficients of various MPU/EUG composites in the frequency range of 4.5–8 kHz were calculated and are displayed in Figure 7. Figure 7 shows how the sound absorption coefficient of all MPU/EUG composites increased first and then decreased with the increase in pressure. Under the pressures of 1.5 MPa and 2 MPa, all MPU/EUG composites exhibited relatively good sound absorption performances. Furthermore, adding EUG into MPU enhanced the sound absorption performances of various MPU/EUG composites. It should be noted that compared with that of pure MPU, the sound absorption coefficient of the MPU/EUG (70/30) composite increased 52.2% at a pressure of 0.1 MPa and 16.8% at a pressure of 4 MPa. The results indicate that EUG significantly improved the sound absorption performances of MPU/EUG composites and that the MPU/EUG (70/30) composite showed the best sound absorption performance in all MPU/EUG composites under both low and high pressures.

### 3.5. Mechanical Properties

The mechanical properties of various MPU/EUG composites were investigated and the relevant data are shown in Table 3 and Figure 8. It can be seen in Table 3 that the hardness and stress at definite elongation (100% and 300%) of various MPU/EUG composites increased gradually with the increase of EUG content. This was mainly due to the fact that EUG can crystallize at room temperature and thus displayed high modulus, which improved the hardness of various MPU/EUG composites and enhanced their ability to resist external deformation [17,46,47]. Additionally, with an increase in the content of MPU, the tensile strength of various MPU/EUG composites decreased first and then increased slightly. The tensile strength of pure EUG is lower than that of pure MPU, so the addition of EUG could reduce the tensile strength of MPU/EUG composites. Specifically speaking, in the tensile process of pure MPU, the molecular chains were oriented and manifested high tensile strength. With the addition of EUG, the existence of EUG crystals can hinder the orientation of the molecular chains and reduce the tensile strength of various MPU/EUG composites [10]. However, when the proportion of EUG in MPU/EUG was greater, the EUG changed from a dispersed phase to a continuous phase, leading to an increase in tensile strength. It is worth noting that the tensile strength and elongation at break of the MPU/EUG (70/30) composite were 25.2 MPa and 701% respectively, which meets the mechanical property requirements of sound absorbing materials.

As can be seen from the stress–strain curves in Figure 8, the mechanical properties of EUG maintained the characteristics of a plastic at room temperature. With the gradual addition of EUG, the prepared MPU/EUG composites underwent a transition from soft to hard (elastomer to plastic) and the modulus of various MPU/EUG composites increased gradually. Adding a small amount of EUG into MPU did not change the tensile curve tendency, which was similar to the tensile curve of pure MPU. With a further increase of EUG, the yield and high modulus of plastic properties appeared on the stress–strain curves of various MPU/EUG composites. Similar phenomena can also be observed in the literature [38,41].

## 4. Conclusions

In this paper, various MPU/EUG composites were successfully fabricated via physical blending methods. With the increase of EUG content, the microstructure surface of various MPU/EUG composites became noticeably rough and cracked, the E’ increased in the temperature range of −50 °C to 40 °C and the tanδ gradually decreased. Increasing the EUG content also meant that the hardness and stress at the definite elongation of various MPU/EUG composites increased gradually, and the tensile strength and elongation at break of the MPU/EUG (70/30) composite were 25.2 MPa and 701% respectively, which meets the mechanical property requirement of sound absorbing materials. Importantly, the incorporation of EUG dramatically enhanced the sound absorption performances of MPU in the frequency range of 4.5 kHz–8 kHz. Compared with that of pure MPU, the sound absorption coefficient of the MPU/EUG (70/30) composite increased 52.2% at a pressure of 0.1 MPa and 16.8% at a pressure of 4 MPa. This research offers a promising strategy for the development of high performing sound absorption materials.

## Figures and Tables

**Figure 1 materials-14-07487-f001:**
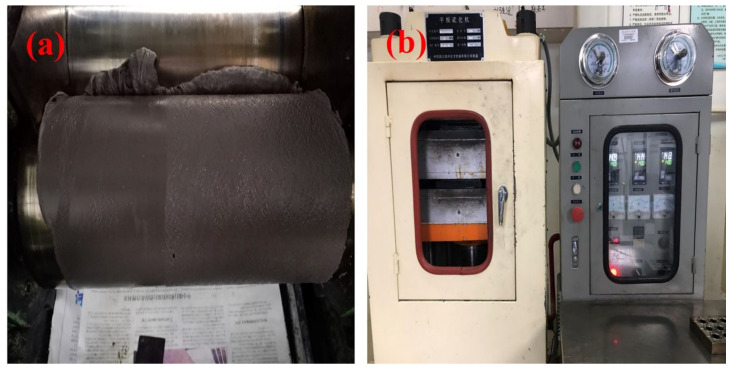
(**a**) the preparation process of various MPU/EUG composites by using an open mill; (**b**) the vulcanizing instrument used in this work.

**Figure 2 materials-14-07487-f002:**
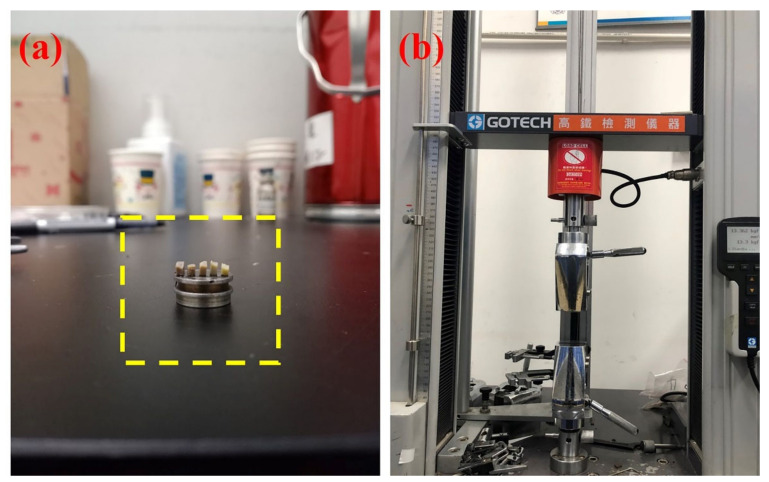
(**a**) the prepared MPU/EUG specimens for SEM observation; (**b**) mechanical property testing instrument used in this work.

**Figure 3 materials-14-07487-f003:**
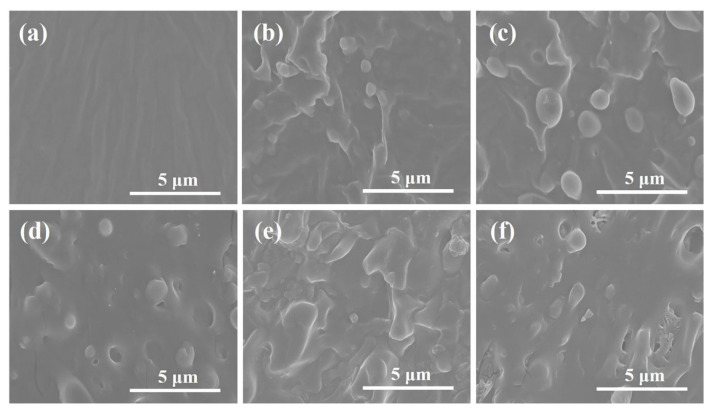
Scanning electron microscope images of various MPU/EUG composites: (**a**) pure MPU; (**b**) MPU/EUG = 90/10; (**c**) MPU/EUG = 80/20; (**d**) MPU/EUG = 70/30; (**e**) MPU/EUG = 60/40; (**f**) MPU/EUG = 50/50.

**Figure 4 materials-14-07487-f004:**
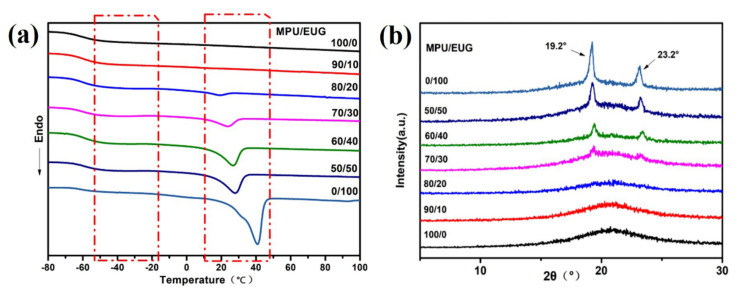
DSC and XRD curves of various MPU/EUG composites. (**a**) DSC curves; (**b**) XRD curves.

**Figure 5 materials-14-07487-f005:**
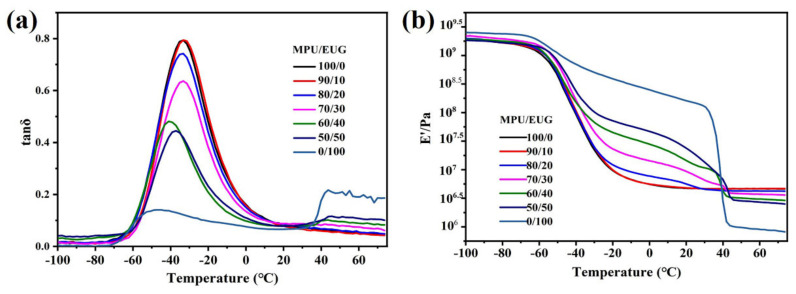
The tanδ-T and E’-T curves of various MPU/EUG composites. (**a**) tanδ-T curves; (**b**) E’-T curves.

**Figure 6 materials-14-07487-f006:**
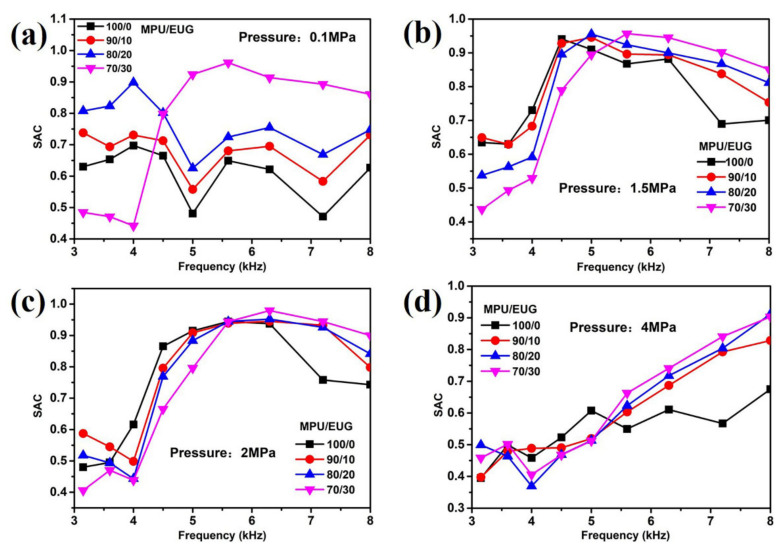
The sound absorption coefficient of various MPU/EUG composites under different pressures: (**a**) 0.1 MPa; (**b**) 1.5 MPa; (**c**) 2 MPa; (**d**) 4 MPa.

**Figure 7 materials-14-07487-f007:**
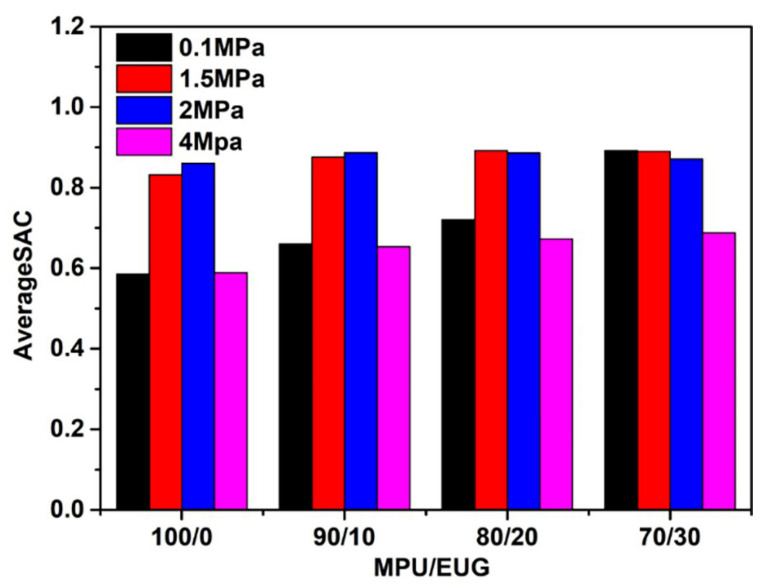
The average sound absorption coefficient of various MPU/EUG composites in the frequency domain of 4.5–8 kHz.

**Figure 8 materials-14-07487-f008:**
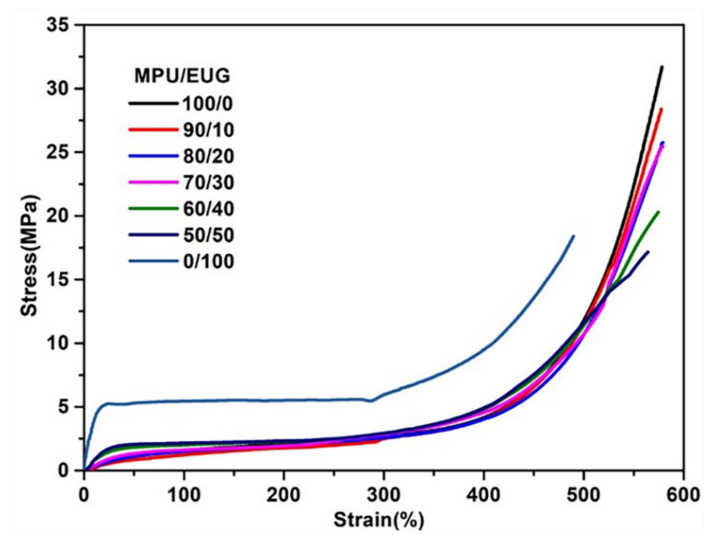
Stress–strain curves of various MPU/EUG composites.

**Table 1 materials-14-07487-t001:** Experimental formula of various MPU/EUG composites.

Component	Content	Component	Content
MPU	100	EUG	100
Stearic acid	0.5	ZnO	5
Active agent NH-2	2	Stearic acid	2
Accelerator D	2	Accelerator DZ	1
Accelerator DM	2	Sulfur	2
Sulfur	2		

**Table 2 materials-14-07487-t002:** Crystallinity of various MPU/EUG composites.

MPU/EUG	100/0	90/10	80/20	70/30	60/40	50/50	0/100
ΔH (j.g^−1^)	0	0	6.35	17.33	25.08	25.22	29.76
*X_c_* (%)	0	0	3.4	9.3	13.4	13.5	15.9

**Table 3 materials-14-07487-t003:** Mechanical properties of various MPU/EUG composites.

MPU/EUG	Tensile Strength (MPa)	Elongation at Break (%)	100% Elongation Stress (MPa)	300% Elongation Stress (MPa)	Hardness (Shao A)	Tensile Permanent Deformation (%)
100/0	30.2	597	1.6	2.7	56	12
90/10	28.5	585	1.5	2.7	55	17
80/20	25.5	609	1.5	2.5	54	26
70/30	25.2	701	1.7	2.5	59	34
60/40	20.9	726	1.9	2.9	65	67
50/50	16.9	679	2.1	3.4	69	130
0/100	17.8	514	5.5	9.3	90	--

## Data Availability

Data sharing is not applicable to this article.

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
