# Peer review of "Fabrication of Millable Polyurethane Elastomer/Eucommia Ulmoides Rubber Composites with Superior Sound Absorption Performance"

_materials, 2021, doi:10.3390/ma14237487_

Round 1

Reviewer 1 Report

Describe better how you prepared the specimen, what and exactly what you are measuring.
Better describe how you performed the measurements of the characteristics of the material, the description is too limited.
You must insert photos of the equipment you used.
You have to insert the photos of the test pieces under tests.
I have seen fig. 4, the acoustic characteristics are high frequency (above 3KHz). Where and how can this material be used?
Fig. 4 Insert letters (A, B, C, D) to better understand which material you are measuring. The first fig. 4 the data is very variable, you can explain that, but other measures are also variable.
Read the papers by Iannace, Ciaburro who performed measurements on rubber.
Improve the quality of Fig. 6.
Of all the measurements you have made, discuss the results and expand the discussion, then compare your results with measurements of similar material.

Reviewer 2 Report

The authors described the process of producing samples and the way of their characterization correctly.

The authors did not describe the polyurethane they used in the research.

Already in the abstract, there are signs E 'and tan delta - without explaining their meaning.

A platform of highly elastic state appears on the E 'curves, which makes it possible to calculate the cross-linking density of the analyzed material. Its calculation will allow quantifying changes in the tested materials.

It's a good idea to increase the size of the images in Figure 1.

Round 2

Reviewer 1 Report

improve quality of images

Reviewer 2 Report

It is advisable that in subsequent works the authors attempt
to determine the degree of cross-linking on the basis of the results
of the DMA analysis.
